# Genome-Wide Association Studies for Methane Production in Dairy Cattle

**DOI:** 10.3390/genes10120995

**Published:** 2019-12-02

**Authors:** R. Calderón-Chagoya, J. H. Hernandez-Medrano, F. J. Ruiz-López, A. Garcia-Ruiz, V. E. Vega-Murillo, M. Montano-Bermudez, M. E. Arechavaleta-Velasco, E. Gonzalez-Padilla, E. I. Mejia-Melchor, N. Saunders, J. A. Bonilla-Cardenas, P. C. Garnsworthy, S. I. Román-Ponce

**Affiliations:** 1Instituto Nacional de Investigaciones Forestales, Centro Nacional de Investigación Disciplinaria en Fisiología y Mejoramiento Animal, Agrícolas y Pecuaria, SADER, Querétaro 76230, Mexico; chagoya.rene@inifap.gob.mx (R.C.-C.); garcia.adriana@inifap.gob.mx (A.G.-R.); montano.moises@inifap.gob.mx (M.M.-B.);; 2Facultad de Medicina Veterinaria y Zootecnia, Universidad Nacional Autónoma de México, Av. Universidad 300, Ciudad de México 04510, Mexicoever@unam.mx (E.G.-P.);; 3School of Biosciences, The University of Nottingham, Sutton Bonington Campus, Loughborough LE12 5RD, UK; neil.saunders@nottingham.ac.uk (N.S.);; 4Campo Experimental La Posta, Centro de Investigación Regional Golfo-Centro, Instituto Nacional de Investigaciones Forestales, Agrícolas y Pecuarias, SADER, Veracruz 94277, Mexico; vega.vicente@inifap.gob.mx; 5Campo Experimental Santiago-Ixcuintla, Centro de Investigación Regional Pacifico-Centro, Instituto Nacional de Investigaciones Forestales, Agrícolas y Pecuarias, SADER, Nayarit 63300, Mexico; bonilla.jorge@inifap.gob.mx; 6Red de Investigación e Innovación Tecnológica para la Ganadería Bovina Tropical (REDGATRO), National Autonomous University of Mexico, Ciudad de México 04510, Mexico

**Keywords:** methane production, dairy cattle, GWAS, SNP, milk yield

## Abstract

Genomic selection has been proposed for the mitigation of methane (CH_4_) emissions by cattle because there is considerable variability in CH_4_ emissions between individuals fed on the same diet. The genome-wide association study (GWAS) represents an important tool for the detection of candidate genes, haplotypes or single nucleotide polymorphisms (SNP) markers related to characteristics of economic interest. The present study included information for 280 cows in three dairy production systems in Mexico: 1) Dual Purpose (*n* = 100), 2) Specialized Tropical Dairy (*n* = 76), 3) Familiar Production System (*n* = 104). Concentrations of CH_4_ in a breath of individual cows at the time of milking (MEIm) were estimated through a system of infrared sensors. After quality control analyses, 21,958 SNPs were included. Associations of markers were made using a linear regression model, corrected with principal component analyses. In total, 46 SNPs were identified as significant for CH_4_ production. Several SNPs associated with CH_4_ production were found at regions previously described for quantitative trait loci of composition characteristics of meat, milk fatty acids and characteristics related to feed intake. It was concluded that the SNPs identified could be used in genomic selection programs in developing countries and combined with other datasets for global selection.

## 1. Introduction

Increase in demand for animal products, particularly meat and milk [1], has stimulated producers to improve the efficiency of production systems and reduce their environmental costs due to consumer concern about the effects of anthropogenic activities on global warming. Figures from FAO considering land use, processing and transportation for the final product, indicate that the livestock sector contributes nearly 18% of the total anthropogenic greenhouse gas (GHG) emissions [2]. However, the dairy industry is believed to be responsible for approximately 4% of global GHG emissions [2], with CH_4_ being one of the main GHGs emitted from this industry. CH_4_ is colorless and odorless with a global warming potential 25 times higher than carbon dioxide (CO2) [3].

In dairy cattle, many strategies have been proposed to mitigate CH_4_, including the reduction of herd size, increased milk yield, diet manipulation, use of CH_4_ inhibitors, immunization against methanogenic archaea, and direct and indirect selection for CH_4_ emission in a breeding program [4]. Research in cattle and sheep has described considerable variability in CH_4_ emissions between individuals fed on the same diet [5], which seems to be related to rumen microbial populations and ruminal kinetics and is also influenced by salivation, rumination and feeding behaviors [6]. Studies on the stoichiometry between volatile fatty acids (VFA) and CH_4_ [7] indicated that CH_4_ is synthesized by microorganisms (Archea being the most important) from compounds released during VFA production, mainly from substrates such as H_2_ and CO_2_ in addition to formate, methylamines, and methanol from demethylation of plant polymers [8]. Moreover, short-chain fatty acids are precursors of milk fatty acids [9]; therefore, a high association between CH_4_ production and milk composition has been suggested [10].

The genome-wide association study (GWAS) represents an important tool for the detection of candidate genes, haplotypes or single nucleotide polymorphisms (SNPs) related to characteristics of economic interest [11]. Advances in genomic selection have proven to aid dairy cattle selection to improve the genetic gain of several characteristics, especially those with low heritability, such as genomic regions associated with CH4 emissions [12]. 

The heritability of CH_4_ emission has been reported to be low [13], which makes a selection for this characteristic difficult. Moreover, information regarding CH_4_ emission from animals in different environments and production systems in developing countries is scarce. Consequently, the scope for breeding animals with low CH_4_ emissions in developing countries is unknown, even though these countries account for the majority of cattle in the world [14]. Cattle production in developing countries often features local *Bos taurus indicus* breeds and crosses, and reliance on locally grown forage crops. Therefore, the genetic markers for CH_4_ emissions established for *Bos taurus taurus* cattle in developed countries might not be applicable in developing countries. The implicit scope of the present study was to investigate characteristics that could be associated with CH_4_ production in order to indirectly select against CH_4_ production in the future. Therefore, the aim of the present study was to identify genomic regions associated with CH_4_ emissions by dairy cattle in temperate and tropical areas. The hypothesis of this project was that by using CH_4_ measurements in three dairy systems, general markers for characteristics associated with CH_4_ production would be identified.

## 2. Materials and Methods 

A total of 280 cows kept in 10 commercial cattle dairy farms with three different production systems across Mexico were used: 1) Dual Purpose (DP, *n* = 100, herds = A, B, C), composed of *Bos taurus taurus* (Simmental, Holstein or Brown Swiss) and *Bos taurus indicus* crosses (Zebu); 2) Familiar Production System (FS, *n* = 104, herds = D, E, F, G), composed mainly of Holstein cattle and finally, 3) Specialized Tropical Dairy (ST, *n* = 76, herds = H, I), mainly Holstein, Brown Swiss, and their crosses. 

Feeding in the DP and ST systems consisted of grazing forage grasses such as Star of Africa (*Cynodon plectostachyus*), Tanzania and Mombaza (*Megathyrsus maximus*), Pangola (*Digitaria decumbens*) and *Brachiaria* spp. Supplemented with concentrates containing 200 g CP/kg at the time of milking. 

Samples of commercial concentrates, complete diets, energy and protein ingredients, and forage grasses were collected from each herd. The samples were sent to the CENID FyMA nutrition laboratory and their components were determined with the methodologies of Van Soest et al. [15] and Weiss et al. [16]. The average percentage composition of the pastures through the study was Dry Matter (DM) 35.1, Crude Protein (CP) 8.0, Neutral Detergent Fiber (NDF) 70.4, Acid Detergent Fiber (ADF) 39.8, lignin 5.7. The estimated consumption of dry matter (DMI) was on average 12.0 kg per cow per day (Table 1).

Feeding in the FS consisted of a total mixed ration (TMR) or grazing plus supplementation with concentrates at time of milking moment (Table 1). The average percentage composition of the diet was DM 65.0, CP 15.5, NDF 39.4, ADF 22.8, Lignin 2.8 and Ash 7.6. The ratio of forage:concentrate was on average 39:61. Using a tape measure, the average weight of the cows was 546 kg. The average DMI was estimated at 17.4 kg per cow per day.

### 2.1. Measurement of CH_4_ Emissions

CH_4_ emissions were estimated for individual cows during milking using an infrared sensor system, (Guardian NG, Infrared Gas Monitor), calibrated and used according to Garnsworthy et al. [17]. During milking and for six weeks, each cow was semi-restrained with the head enclosed in a Perspex head box. Cows were offered concentrate individually. A sampling tube was placed in each head box, through which air was sampled at the rate of 2 liters per minute. Concentrations of CH_4_ were measured continuously. These measurements were used to calculate the emission of CH_4_ per minute per liter of air. The feeders were designed to minimize air currents inside the Perspex head boxes and to obtain reliable measurements without altering daily milking routines. CH_4_ estimations are reported as mg CH_4_/L of air sampled at the time of milking (MEIm) [17].

### 2.2. DNA Sampling and Genotyping

Tail hair, including the hair follicles, was sampled from all the animals in the study. Samples were individually identified and sent to GeneSeek Laboratory (Lincoln, NE, USA) for DNA extraction and genotyping using high-density panels. Two different high-density chips were used for genotyping due to breed variability (both with the reference genome Bos_taurus_UMD_3.1.1): the GGP Bovine LD V4 chip with 30,125 SNPs was used for FS (only Holstein), and the GGP Bovine 150k chip with 138,962 SNPs was used for DP and ST (high number of crossed-bred animals).

### 2.3. Quality Control of Genotypes

Only SNPs common to both chips (*n* = 26,145 SNPs) were used for the genomic analysis. Prior to GWAS, a quality control analysis of genotypes was carried out, in which all SNPs that had a call rate of less than 0.95, a minor allele frequency of less than 0.05 and those that failed to be in Hardy-Weinberg equilibrium in the experimental population (*p* < 0.0001) were not included. Individual animals with a call rate of less than 0.95 were also eliminated. After the quality control analysis, 21,958 autosomal and mitochondrial SNPs from 280 cows were used to carry out genotype association tests. Subsequently, a genotype association test was carried out, including a false positive detection test and Bonferroni test.

### 2.4. Statistical Analysis

CH_4_ production during milking was evaluated using a completely randomized factorial design, with the MIXED procedure using SAS software [18]. The model used was
y_ijk_ = μ + H_i_ + M_j_ + ε_(ij)k_(1)
where*y_ijk_* = MEIm in the k-th observation in the i-th herd and j-th months in milk.*μ* = overall mean.*H_i_ =* fixed effect of the i-th herd.M_j_ = fixed effect of the j-th month in milk.ε_(ij)k_ = CH_4_ production during milking residual in the k-th observation in the i-th herd and j-th months in milk.

The effects of days in milk, average milk yield and dairy system were also evaluated in the model, however, herd and months in milk fit the model significantly better.

The effects included in the model explain part of the environmental variation. The residuals represent the proportion of the variance not explained by the model effects, including genetic variance [19], for that reason, residuals values were used as phenotypes in the GWAS.

### 2.5. Complete Genome Association

A principal component analysis (PCA) was applied to the genotype data to infer continuous axes of genetic variation. Let *g_ij_* be a matrix of genotypes for SNP *i* and individual *j*, where i = 1 to M and j = 1to N. We subtract the row mean *μ_i_* = (∑*_j_ g_ij_*)/N from each entry in row *i* to obtain a matrix with row sums equal to 0. We then normalize row i by dividing each entry by $\sqrt{p_{i}(1-\;p_{i}})$, where *p_i_* is a posterior estimate of the unobserved underlying allele frequency of SNP *i* defined by *p_i_* = (1 + ∑*_j_ g_ij_*)/(2 + 2 N). The resulting matrix is X. We compute an N × N covariance matrix Ψ of individuals, where Ψ*jj*´is defined to be the covariance of column *j* and column *j´* of X. The *k*th axis of variation is the *k*th eigenvector of Ψ. The axes of variation reduce the data to a small number of dimensions, describing as much variability as possible; they are defined as the top eigenvectors of a covariance matrix between samples. This method allows the detection and correction of population stratification on a genome-wide scale, maximizing the detection of true associations and minimizing erroneous associations [20]. Quality control analyses and GWAS were conducted with the Golden Helix program [21]. SNP & Variation Suite incorporates advanced regression technologies that enable one to perform linear and logistic regression, stepwise regression (both backward elimination and forward selection), gene by environment interaction regression, and permutation tests with numeric variables and recorded genotypes.

We use residuals from Equation (1), which are now adjusted for polygenic covariation and fixed effects, as a novel quantitative trait for association analyses with each of many markers using classical methods for unrelated individuals (‘‘population-based design’’). These residuals are used as the dependent trait in a simple linear regression for each SNP.
ê_i_ = μ+ kg_i_ + e_i_(2)
where *ê_i_* is the vector of residuals from (1), *μ* is the mean, *g* is the vector of markers corrected by PCA, *k* is the vector of markers effects, and *e* is the vector of random residuals.

## 3. Results

The descriptive statistics of MEIm (Table 2) for the ST system were 0.082 mg/L and for DP and FS systems, the mean of both was 0.062 mg/L; variability was greater in the DP system than in the other production systems. CH_4_ emissions on DP and FS were similar (*P* < 0.05) and lower than ST (*P* < 0.05). Milk yield was different among the three dairy production systems, as presented in Table 2.

Figure 1 shows the box plot for MEIm and MY. It is observed that the distribution of methane production is similar in all the herds, however, in herds A, H and I, there are outliers. It is worth mentioning that these three herds are in the same region and share high connectivity. The MY is explained by observing that the herds with the highest milk production are those that have pure breeds *Bos taurus taurus* while the cattle crossed with *Bos taurus indicus* have the lowest values.

The data were corrected of population stratification on a genome-wide scale. The PCA analysis for the dairy production systems and breeds is shown in Figure 2. Systems DP and ST showed a more dispersed population stratification, in contrast to FS, which was not stratified.

The quantile–quantile plot (Figure 3) did not show large deviations from the null hypothesis, which means that PCA corrected the population stratification on a genome-wide scale structure. In the Q–Q plot, if the observed values correspond to the expected values, all the points are on or near the middle line between the x-axis and the y-axis (null hypothesis: black line in Figure 3). The observed P values are clearly more significant than expected under the null hypothesis. There is no early separation of the expected from the observed, which means that there is not a population stratification [22].

The SNPs associated with genetic variation in MEIm in the combined dataset are shown in Figure 4. GWAS revealed novel loci associated with CH_4_ emission in dairy cattle. Chromosomes 1, 3, 13 and 20 have most SNPs associated with CH_4_ emission with 11, 7, 5 and 4 SNPs, respectively (Appendix A), being that all specific markers are common in the population. 

## 4. Discussion

The CH_4_ emissions in the present study are lower than those published by Garnsworthy et al. [17] and Bell et al. [23] which were recorded in specialized dairy systems with a higher consumption of dry matter and concentrates than in the current study. Although grain-based diets result in a lower production of CH_4_ compared to diets based on forage [24], in this case, differences are likely due to the consumption of dry matter because the farms measured in the current study had animals grazed whereas specialized systems have diets based on conserved forages and concentrate.

The PCA stratification could be the result of crossbreeding, which is predominant in DP and ST systems. The SNPs associated with genetic variation in MEIm (*p* < 0.001) have been associated with characteristics of meat composition, milk fatty acids and characteristics related to feed intake [25,26,27,28,29,30,31]. 

Correcting the genotype through the PCA, we avoid a high the genomic inflation factor. The genomic inflation factor expresses the deviation of the distribution of the observed test statistic compared to the distribution of the expected test statistic. High genomic inflation factors are caused by population stratification, strong linkage disequilibrium (LD) between SNPs, strong association between SNPs and phenotypes, and systematic bias [32].

As trait complexity increases, the number of loci affecting the trait increases along with environmental interactions with an expected decrease in heritability. Conversely, for complex traits, a higher number of loci affect the trait, there is more interaction with the environment, and there is an expected decrease in heritability. For a trait with a low heritability, the threshold value for significance of associating loci with a trait would have low −log10 (*p*-values) [33]. In published studies in pig populations, threshold values for −log10 (*p*-value) ranged from 3.3 to 6, using either no multiple testing correction, a Bonferroni correction, the false discovery rate, or genomic control [32].

### 4.1. Markers Associated with Dairy Traits

SNPs identified to be associated to methane emission in this study on chromosomes 13 (position 47,642,260), 14 (position 62,204,044) and 19 (position 47,747,001) were, in other studies, associated with milk fatty acid composition for C6:0, cis-9-C16:1 percentage [27] and several other fatty acids [26], respectively. Short-chain fatty acids are precursors of fatty acids in milk [9] and CH_4_ is synthesized by archaea from CO2 and H2 released during VFA production [8]. Previous studies have described associations between CH_4_ and fatty acids in milk, finding both positive and negative correlations [34,35], and these relationships can be used to predict CH_4_ emissions [36].

On chromosome 1 (Appendix A), two markers associated with MEIm were located within the QTL associated with the percentage of myristic acid in milk [26]: one in the *SLC9A9* gene and the other in the *LOC104971015* gene. On chromosome 13, a marker associated with MEIm was found within the QTL associated with the percentage of caproic acid in milk [27], on the *SLC23A2* gene (Appendix A). On chromosome 14 (Appendix A), a marker associated with MEIm was found within the QTL associated with the percentage of palmitoleic acid in milk [27]. On chromosome 17 (Appendix A), two markers associated with CH_4_ were found within the QTL associated with fatty acids in milk [27], both on the *TMEM233* gene. On chromosome 19 (Appendix A) a marker associated with CH_4_ was found within two QTL associated with fatty acids in milk [26], on the *MRC2* gene. Finally, on chromosome 20 (Appendix A) there is a QTL associated with fatty acids in milk [26] that has a marker associated with CH_4_ on the *RAI14* gene.

### 4.2. Markers Associated with Beef Traits

Significant markers for MEIm were found in regions previously associated with characteristics of fatty acids in meat. This study found gene regions associated with methane emissions and through these, genes related to the intramuscular fat content (chromosome 5 position 41,431,327) [28] and content of different fatty acids (i.e., chromosome 1 position 84,485,319: trans-6/9-C18:1; and, chromosome 13 position 36,197,476: C22:1 fatty acid and trans-12-C18:1) [29].

Additionally, relationships were found between our markers on chromosome 1 at position 107,388,026 for dry matter intake and on chromosome 14 at position 7,744,264 for residual feed intake [30]. The production of CH_4_ has been correlated with dry matter intake (*r* = 0.46) [37] and residual food intake (*r* = 0.36) [38].

On chromosome 1 (Appendix A) a QTL related to dry matter intake [30] was found, and it can be seen that the QTL peak and the MEIm marker are in the same *PPM1L* gene. In addition, on chromosome 1 (Appendix A), we found a marker associated with MEIm within a QTL linked to the trans-6/9-C18 fatty acid:1 [29]; this marker is found in the *MCF2L2* gene and is very close to the QTL peak. Appendix A shows markers associated with MEIm inside the QTL for daily weight gain [28]; this is a relatively broad QTL, within which we found seven markers associated with MEIm, going through the genes *LOC107132190, LOC104970931, LOC100138913, NAALADL2, MECOM*, and *PPM1L*.

On chromosome 3 (Appendix A), we found a marker associated with MEIm that is found inside the QTL associated with daily weight gain [31]. On chromosome 5 (Appendix A), we found a marker associated with MEIm within the QTL for intramuscular fat [25], on the *SLC2A13* gene. On chromosome 13, we found a marker associated with MEIm, that is, in the QTL related to the content of the fatty acids C22:1 and trans-12-C18:1 [29], this is very close to the QTL peak (Appendix A).

Finally, our GWAS results concur with those previous observations of markers on chromosomes 4 and 20, associated with CH_4_ production per unit of dry matter intake [13].

The primary aim of this study was to identify genomic regions associated with CH_4_ emissions by dairy and dual-purpose cattle in temperate and tropical areas of Mexico. This aim was achieved and the SNPs identified to provide a solid foundation on which to develop further studies leading to genomic selection for low CH_4_ emissions in developing countries. Of course, selection for low CH_4_ emissions must be in parallel with selection for feed efficiency and other traits of economic importance. The secondary aim of this study was to compare the SNPs identified to the markers reported to be associated with production traits previously related to CH_4_ emissions. This aim was also successful, and the associations discovered provide confidence that the results of this study concur with known biological pathways. Furthermore, the associations provide confidence that the results could be combined with datasets generated in developed countries to provide tools for genomic selection on a global basis.

## 5. Conclusions

In this study, 46 SNPs that have significant associations with MEIm were identified. Several SNPs associated with MEIm were found at regions previously described for QTLs associated with composition characteristics of the meat, milk fatty acids, and characteristics related to feeding intake, such as residual feed intake. Relationships between mammary synthesis of fatty acids and ruminal synthesis of CH_4_ are expected, and correlations between CH_4_ and the fatty acids in milk have been reported, so similar relationships might be expected between CH_4_ and the fatty acids in meat. The SNPs identified in this study may be incorporated into genomic selection programs for low CH_4_ emissions in developing countries and could be combined with other datasets to provide tools for genomic selection on a global basis. A future direction for this work could be to sequence the specific regions where the significant markers are found and compare between breeds. This could give stronger results for the characteristics associated with methane production.

## Figures and Tables

**Figure 1 genes-10-00995-f001:**
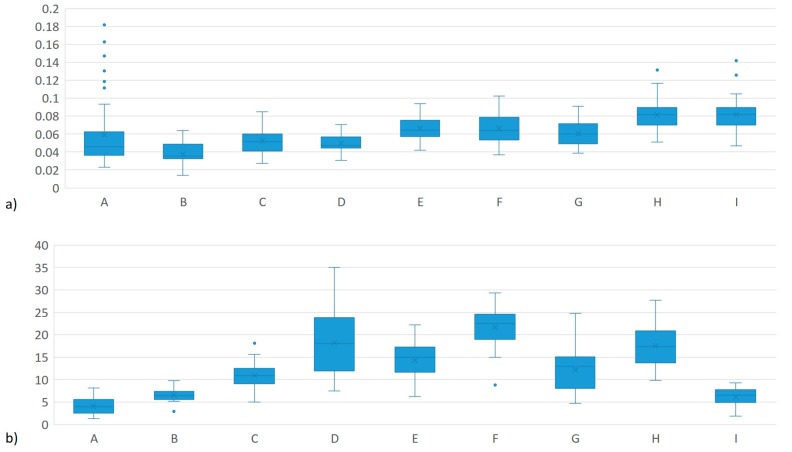
Box plot for **a**) mg of CH_4_/L of air sampled at the time of milking (MEIm) and **b**) milk yield (MY) in three production systems in Mexico.

**Figure 2 genes-10-00995-f002:**
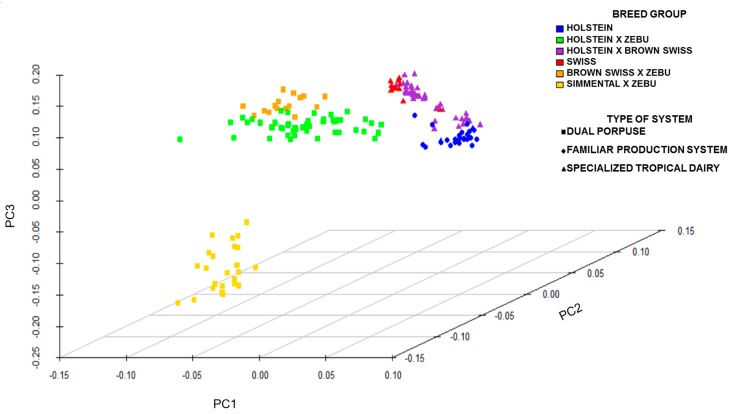
Principal Components Analysis (PCA) for the breed and dairy production system.

**Figure 3 genes-10-00995-f003:**
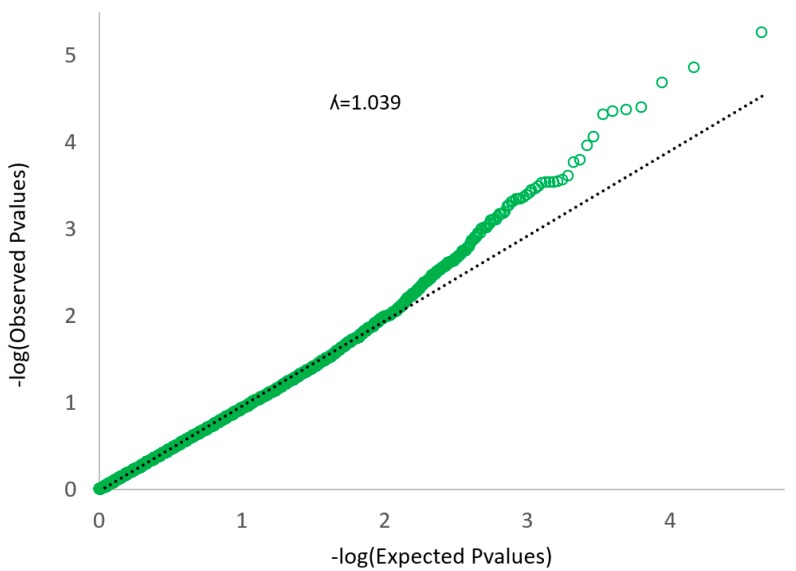
Quantile–quantile (Q–Q) plot of the data shown in the Manhattan plot.

**Figure 4 genes-10-00995-f004:**
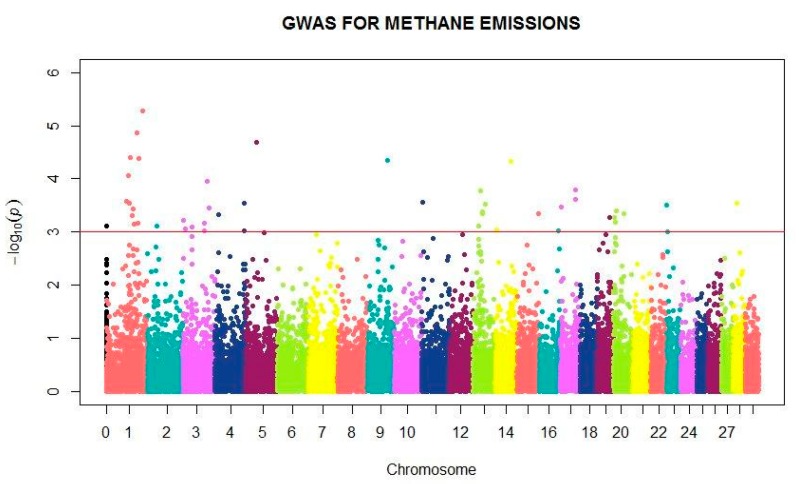
Manhattan plot showing the main SNPs within autosomal and mitochondrial DNA (chromosome 0) associated with methane production during milking in dairy cattle.

**Table 1 genes-10-00995-t001:** Feeding by the dairy herd.

Feeding	Herds								
	A	B	C	D	E	F	G	H	I
Concentrate (kg)	3.5	2	2	11	6.6	10.4	6.3	3.5	3.5
Corn silage (kg)						5.2			
Alfalfa hay (kg)						1.7			
Corn stubble (kg)				5.4	8	4.8	3.3		
*Saccharum sinense* (kg)	20							20	20
***Grazing***									
*Cynodon plectostachyus Vanderyst* (kg)	X	X						X	X
*Sorghum vulgare* (kg)		X							
*Megathyrsus maximus* (kg)		X	X						
*Brachiaria decumbens Stapf* (kg)		X	X						
*Desmodium ovalifolium* (kg)		X							
*Andropogon gayanus Kunth* (kg)		X							
*Pennisetum sp.* (kg)			X						
*Digitaría decumbens Stend* (kg)			X						
Native grasses				X	X	X	X		

**Table 2 genes-10-00995-t002:** Descriptive statistics of mg of CH_4_/L of air sampled at the time of milking (MEIm) and milk yield (MY) in three production systems in Mexico.

System	Herd	*n*	MEIm (mg/L)		MY (kg per day)	
			*µ*	*σ*	*µ*	*σ*
Dual Purpose	A	51	0.076 ^ab^	0.076	4.2 ^g^	2.0
	B	16	0.038 ^bc^	0.012	6.5 ^g^	1.6
	C	33	0.053 ^abc^	0.015	10.9 ^ef^	2.8
	TOTAL	100	0.062 ^A^	0.057	6.8 ^A^	3.7
Familiar Production System	D	24	0.050 ^abc^	0.010	18.2 ^abc^	7.6
	E	16	0.067 ^abc^	0.013	14.3 ^de^	3.8
	F	33	0.067 ^abc^	0.017	21.6 ^ab^	4.8
	G	31	0.061 ^abc^	0.014	12.1 ^def^	4.4
	TOTAL	104	0.062 ^A^	0.015	15.9 ^C^	6.1
Specialized Tropical Dairy	H	38	0.082 ^a^	0.017	17.5 ^bc^	4.7
	I	38	0.082 ^abc^	0.018	6.1 ^g^	2.0
	TOTAL	76	0.082 ^B^	0.017	11.8 ^B^	6.8

^a, b, c, d, e, f, g^ Values of the herd within a row with different superscripts differ significantly at *P* < 0.05. ^A, B, C^ Values of the system within a row with different superscripts differ significantly at *P* < 0.05.

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
