# Peer review of "Genome-Wide Association Studies for Methane Production in Dairy Cattle"

_genes, 2019, doi:10.3390/genes10120995_

Round 1
Reviewer 1 Report
The paper treats the question of methane emission in cattle and its genomic background. The authors work with different populations and phenotypes of 280 cattle.
Many details in the methodology part and some results are missing, which are essential for the proper understanding and to see if analyses were performed correctly.
My major concern is about the stratification in your data: Extremely different systems with very different cattle breeds are combined. There is no evidence in the paper, whether this huge stratification could have been corrected for sufficiently. Furthermore this point should be discussed in more detail.
Generally the discussion is very detailed about the QTL discovered and their relationship to other QTL. However the discussion on the GWAS methodology, the combination of breeds, but also the experimental data for methane emission should be discussed in more detail.
Please also comment on issues arising from the indicus x taurus crossbreds. Especially in the SNP data it would be interesting to know if these individuals show systematic differences due to the ascertainment bias.
It would also be interesting to have a boxplot with the distribution of the raw phenotypes (within herd for instance).
Have the authors thought of other possibilities to combine the data, such as a within population GWAS and a subsequent meta-analysis?
Furthermore the model describing the phenotype seems strange: According to the description of equation 1 residuals will only be available per herd and not per individual. Also it is not clear from the equations, where the PCA are accounted for. Please explain this procedure more accurately.
Additionally I was wondering why only such a small number of SNPs are used when only for the pure Holstein cattle a low-density chip was used. Wasn't it possible to impute this dataset to 150K as well, which should then lead to a higher number of SNPs.
Then please also specify why you call the animals in this study tropical cattle: most of the breeds used in the study are conventional taurus breeds, that are kept all over the world. From the paper it is unclear where (country/region) the data has been collected. The number of herds per system also only becomes evident in the tables 1 and 2.
For the GWAS the threshold is extremely low. Please justify this threshold.
Please specify on which genome assembly you are working on. According to a quick check I made on 2 SNPs it looks like that you are working on UMD3.1 is this correct? If this assumption is correct, then why are you not using the new reference genome ARS-UCD1.2?
Generally it seems that the results part is extremely short.
Table 1: How were the components for grazing determined?
Table 2: Please specify the unit for the MY (kg per day?)
Figure 1: What does the EV = ... mean? Please name the axis properly. The same for the Breed designation: What is "SWISS"?
Figure 2: Why is there a chromosome 0 plotted?
Author Response
Answer to the Academic Editor Notes.
The authors appreciate the comments. In general all the comments has been addressed.
REVIEWER 1.
English language and style are fine/minor spell check required.
The paper treats the question of methane emission in cattle and its genomic background. The authors work with different populations and phenotypes of 280 cattle.
Many details in the methodology part and some results are missing, which are essential for the proper understanding and to see if analyses were performed correctly.
My major concern is about the stratification in your data: Extremely different systems with very different cattle breeds are combined. There is no evidence in the paper, whether this huge stratification could have been corrected for sufficiently. Furthermore this point should be discussed in more detail.
LINE 142-147: “A principal component analysis (PCA) was applied to the genotype data to infer continuous axes of genetic variation. The axes of variation reduce the data to a small number of dimensions, describing as much variability as possible; they are defined as the top eigenvectors of a covariance matrix between samples. This method allows the detection and correction of population stratification on a genome-wide scale, maximizing detection of true associations and minimizing erroneous associations” DONE
LINE 171-172: “Supported by the PCA, the data were corrected of population stratification on a genome-wide scale.” DONE
Generally the discussion is very detailed about the QTL discovered and their relationship to other QTL. However the discussion on the GWAS methodology, the combination of breeds, but also the experimental data for methane emission should be discussed in more detail.
DONE
Please also comment on issues arising from the indicus x taurus crossbreds. Especially in the SNP data it would be interesting to know if these individuals show systematic differences due to the ascertainment bias.
Although that would be a very promising goal, we are limited by our number of observations. DONE
It would also be interesting to have a boxplot with the distribution of the raw phenotypes (within herd for instance).
LINES 168-172: Figure 1 shows the box plot for MEIm and MY. It is observed that distribution of methane production is similar in all herds, however in herds A, H and I there are outliers, it is worth mentioning that these three herds are in the same region and share high connectivity. The MY is explained by observing that the herds with the highest milk production are those that have pure breeds Bos taurus taurus while the cattle crossed with Bos taurus indicus have the lowest values.
Figure 1. Box plot for mg of CH4 / L of air sampled at the time of milking (MEIm) and milk yield (MY) in three production systems in Mexico.
Have the authors thought of other possibilities to combine the data, such as a within population GWAS and a subsequent meta-analysis?
No, for the number of observations we can have false positives/spurious associations DONE
Furthermore the model describing the phenotype seems strange: According to the description of equation 1 residuals will only be available per herd and not per individual. Also it is not clear from the equations, where the PCA are accounted for. Please explain this procedure more accurately.
LINE 131: “yijk = MEIm in the k-th observation in the i-th herd and j-th months in milk.”DONE
LINE 142-147: “A principal component analysis (PCA) was applied to the genotype data to infer continuous axes of genetic variation. The axes of variation reduce the data to a small number of dimensions, describing as much variability as possible; they are defined as the top eigenvectors of a covariance matrix between samples. This method allows the detection and correction of population stratification on a genome-wide scale, maximizing detection of true associations and minimizing erroneous associations”DONE
Additionally I was wondering why only such a small number of SNPs are used when only for the pure Holstein cattle a low-density chip was used. Wasn't it possible to impute this dataset to 150K as well, which should then lead to a higher number of SNPs.
That may have been possible, but as we work with commercial dairy cattle, they did not have in most cases genealogical information about the animals, so we could not be 100% sure that they were pure Holstein. DONE
Then please also specify why you call the animals in this study tropical cattle: most of the breeds used in the study are conventional taurus breeds, that are kept all over the world. From the paper it is unclear where (country/region) the data has been collected. The number of herds per system also only becomes evident in the tables 1 and 2.
LINE 78-83: “A total of 280 cows kept in 10 commercial cattle dairy farms with three different production systems across Mexico were used: 1) Dual Purpose (DP, n = 100, herds = A, B, C), composed of Bos taurus taurus (Simmental, Holstein or Brown Swiss) and Bos taurus indicus crosses (Zebu); 2) Specialized Tropical Dairy (ST, n = 76, herds = H, I), mainly Holstein, Brown Swiss, and their crosses; and finally, 3) Familiar Production System (FS, n = 104, herds= D, E, F, G), composed mainly of Holstein cattle.” DONE
I mention in the manuscript the "Specialized Tropical Dairy", but this is the translated name by which the system is known in the region
For the GWAS the threshold is extremely low. Please justify this threshold.
LINE 175-177: “A limitation for the GWAS is the small sample size, which is why the threshold was lowered to a significance of p<0.001”
Please specify on which genome assembly you are working on. According to a quick check I made on 2 SNPs it looks like that you are working on UMD3.1 is this correct? If this assumption is correct, then why are you not using the new reference genome ARS-UCD1.2?
LINE 114: “(both with the reference genome Bos_taurus_UMD_3.1.1)”. About the reference genome ARS-UCD1.2, by the time we genotype the animals and elaborated the GWAS, there wasn´t released that reference genome (between 2015-2017)
Generally it seems that the results part is extremely short.
DONE
Table 1: How were the components for grazing determined?
LINES 88-91: “Samples of commercial concentrates, complete diets, energy and protein ingredients, and forage grasses were collected from each herd. The samples were sent to the CENID FyMA nutrition laboratory and its components were determined with the methodologies of Van Soest et al. [14] and Weiss et al. [15].” DONE
Table 2: Please specify the unit for the MY (kg per day?)
Table 2. Kg per day DONE
Figure 1: What does the EV = ... mean? Please name the axis properly. The same for the Breed designation: What is "SWISS"?
EV was the eigenvectors. but I better chance EV per PC1-3 and describe more about PCA in the manuscript DONE
Figure 2: Why is there a chromosome 0 plotted?
LINE 122-123: “…21,958 autosomal and mitochondrial SNPs…” DONE

Reviewer 2 Report
The paper by Calderon-Chagoya et al describes a GWAS for methane production in cattle. The authors found 46 variants that were significant with CH4 production in their association analysis. I found the paper to be well written and well described. I have a few questions/concerns.
1. The authors use a significance level for SNPs of 0.001. The standard accepted significance level for GWAS is 5 x 10-8. Why did the authors use this less stringent threshold of 0.001 for significance?
2. For the variants that were found to be significant under the more liberal threshold, what are the minor allele frequencies of these SNPs? Are they common throughout the population or are they rare and being driven by only a few individuals?
3. Please discuss the issue of power in this study, as a small sample size of 280 cows was used.
4. Please expand on the future directions of this work. Is functional analysis planned on the genes of interest? Is targeted sequencing of the associated regions planned?
Author Response
Answer to the Academic Editor Notes.
The authors appreciate the comments. In general all the comments has been addressed.
REVIEWER 2.
English language and style are fine/minor spell check required
The paper by Calderon-Chagoya et al describes a GWAS for methane production in cattle. The authors found 46 variants that were significant with CH4 production in their association analysis. I found the paper to be well written and well described. I have a few questions/concerns.
The authors use a significance level for SNPs of 0.001. The standard accepted significance level for GWAS is 5 x 10-8. Why did the authors use this less stringent threshold of 0.001 for significance?
LINE 175-177: “A limitation for the GWAS is the small sample size, which is why the threshold was lowered to a significance of p<0.001”
For the variants that were found to be significant under the more liberal threshold, what are the minor allele frequencies of these SNPs? Are they common throughout the population or are they rare and being driven by only a few individuals?
Table S1 MAF
LINE 120: “a minor allele frequency of less than 0.05”
LINE 182: “…being that all specific markers are common in the population”.
Please discuss the issue of power in this study, as a small sample size of 280 cows was used.
LINE 175-177: “A limitation for the GWAS is the small sample size, which is why the threshold was lowered to a significance of p<0.001”
Please expand on the future directions of this work. Is functional analysis planned on the genes of interest? Is targeted sequencing of the associated regions planned?
LINE 252-256: “The SNPs identified in this study may be incorporated into genomic selection programs for low CH4 emissions in developing countries and could be combined with other datasets to provide tools for genomic selection on a global basis. A future direction for this work could be the sequence the specific regions where the significant markers are found and compare between breeds. This could give stronger results for the characteristics associated with methane production.” DONE

Round 2
Reviewer 1 Report
The authors addressed all questions and the paper improved substantially from the first version. However many comments are only partially adressed and the issues mentioned not always solved in the current version of the paper.
To me the major issue is still that there is no evidence, that they could correct properly for the huge stratification. So please provide qq-plots or/and lambda values. I don't see how the PCA plot shows that the stratification could have been corrected for.
Then I have some comments to the reply letter:
Reply letter L21-25: Where is this done?
Reply letter L26-30: This actually goes hand in hand with the correction of the stratification: There might be some differences in the populations due to the inclusion of indicus samples and I would like to know how the authors dealt with this.
Reply letter 42-45: You can also have spurious observations and false positives if your PCA doesn't correct sufficiently
Reply letter L52-57: This still doesn't explain wher in your modell the PCA is accounted for
Reply letter L58-63: Why it should not be possible to impute to a higher density by a population-based approach due to the uncertainty of the breed designation?
Reply letter L121-124: Simply saying that the threshold was low due to the small sample size is not really a discussion about power issues.
Comments directly to the revised version of the paper:
L97-101: The naming of the herds already here is helpful for the understanding, however the order is a little bit confusing because the herds in the middle are H,I and the last ones are D-G, it would be nice if the herds would be described in the order of the names
L112-113: This description is actually quite unclear: Had herds D-G grazing or not? If they had why there is not composition of the pasture in table 1? And what do the 40% and 60% of the herds mean? (Aren't there just these four herds; or was feeding different within herd?
L134: Where is the description of table 1?
L169: How was the covariance matrix between the samples defined?
L173-176: This information doesn't seem relevant to me in that context. I'm not that much interested in the possibilities SVS offers but more in what was used in this study.
L177: It is still not clear to me how the PCA was corrected for, it must be somewhere between equation 1 and here. Was there another step between equation 1 and 2 or were the PCAs included in eq 2?
L206: In the description of Figure 1 please add the a and b names to make clear which plot is what
Figure 3: It should be mentioned somewhere that what is named 0 are the MT SNPs
Author Response
Answer to the Academic Editor Notes.
The authors appreciate the comments. In general all the comments has been addressed.
REVIEWER 1.
The authors addressed all questions and the paper improved substantially from the first version. However many comments are only partially addressed and the issues mentioned not always solved in the current version of the paper.
To me the major issue is still that there is no evidence that they could correct properly for the huge stratification. So please provide qq-plots or/and lambda values. I don't see how the PCA plot shows that the stratification could have been corrected for.
DONE
Figure 3
The quantile–quantile plot (Figure 3) did not show large deviations from the null hypothesis, which means that PCA corrected the population stratification on a genome-wide scale structure. In the Q-Q plot if the observed values correspond to the expected values, all points are on or near the middle line between the x-axis and the y-axis (null hypothesis: black line in Figure 3). The observed P values are clearly more significant than expected under the null hypothesis. There is not an early separation of the expected from the observed, this means, there is not a population stratification [22].
Then I have some comments to the reply letter:
Reply letter L21-25: Where is this done?
I just integrated in this second review more information in the discussion and results part
Reply letter L26-30: This actually goes hand in hand with the correction of the stratification: There might be some differences in the populations due to the inclusion of indicus samples and I would like to know how the authors dealt with this.
This comments if clearly something helpful. Unfortunately, there were not Bos tuarus indicus pure breed genotypes available for this study. Even thought, the inclusion of multibreed Bos taurus taurus and Bos taurus indicus genotypes did a good enough correction with the stratification.
Reply letter 42-45: You can also have spurious observations and false positives if your PCA doesn't correct sufficiently
Reply letter L52-57: This still doesn't explain wher in your modell the PCA is accounted for
“A principal component analysis (PCA) was applied to the genotype data to infer continuous axes of genetic variation. Let gij be a matrix of genotypes for SNP i and individual j,where i = 1 to M and j = 1to N. We subtract the row mean μi=(∑j gij)/N from each entry in row i to obtain a matrix with row sums equal to 0. We then normalize row i by dividing each entry by √(p_i (1- p_i ),where pi is a posterior estimate of the unobserved underlying allele frequency of SNP i defined by pi = (1 + ∑j gij)/(2 + 2N). the resulting matrix is X. We compute an N x N covariance matrix Ψ of individuals, where Ψjj´is defined to be the covariance of column j and column j´ of X. The kth axis of variation is the kth eigenvector of Ψ. The axes of variation reduce the data to a small number of dimensions, describing as much variability as possible; they are defined as the top eigenvectors of a covariance matrix between samples. This method allows the detection and correction of population stratification on a genome-wide scale, maximizing detection of true associations and minimizing erroneous associations [20].”
“Where êi is the vector of residuals from (1), μ is the mean, g is the vector of markers corrected by PCA, k is the vector of markers effects, and e is the vector of random residuals.”
Reply letter L58-63: Why it should not be possible to impute to a higher density by a population-based approach due to the uncertainty of the breed designation?
Because we would be imputing erroneous information, in addition to having a reference genome of Holstein cattle in Mexico
Reply letter L121-124: Simply saying that the threshold was low due to the small sample size is not really a discussion about power issues.
“Correcting the genotype through the PCA we avoid the genomic inflation factor. The genomic inflation factor expresses the deviation of the distribution of the observed test statistic compared to the distribution of the expected test statistic. High genomic inflation factors are caused by population stratification, strong linkage disequilibrium (LD) between SNPs, strong association between SNPs and phenotypes, and systematic bias [32].
As trait complexity increases, the number of loci affecting the trait increases along with environmental interactions with an expected decrease in heritability. Conversely, for complex traits, higher loci affect the trait, there is more interaction with the environment, and there is an expected decrease in heritability. For a trait with a low heritability, the threshold value for significance of associating loci with a trait would have low – log10 (P)-values [33]. In published studies in pig populations, threshold values for −log10 (p‐value) ranged from 3.3 to 6, using either no multiple testing correction, a Bonferroni correction, the false discovery rate, or genomic control [32]”
Comments directly to the revised version of the paper:
L97-101: The naming of the herds already here is helpful for the understanding, however the order is a little bit confusing because the herds in the middle are H,I and the last ones are D-G, it would be nice if the herds would be described in the order of the names
DONE
L112-113: This description is actually quite unclear: Had herds D-G grazing or not? If they had why there is not composition of the pasture in table 1? And what do the 40% and 60% of the herds mean? (Aren't there just these four herds; or was feeding different within herd?
Family systems do not invest much in grazing, so they use only native pastures. As for the 4 herds, one of the herds had two different owners (the father and the son) and each one handled their animals differently, (I deleted 40 and 60% to avoid confusion).
Tabla 1 DONE
“Feeding in the FS consisted of a total mixed ration (TMR), or grazing plus supplementation with concentrates at time of milking moment (Table 1).”
L134: Where is the description of table 1?
I'm not sure if it refers to the title, if that's it, I already fixed it
L169: How was the covariance matrix between the samples defined?
“A principal component analysis (PCA) was applied to the genotype data to infer continuous axes of genetic variation. Let gij be a matrix of genotypes for SNP i and individual j,where i = 1 to M and j = 1to N. We subtract the row mean μi=(∑j gij)/N from each entry in row i to obtain a matrix with row sums equal to 0. We then normalize row i by dividing each entry by √(p_i (1- p_i ),where pi is a posterior estimate of the unobserved underlying allele frequency of SNP i defined by pi = (1 + ∑j gij)/(2 + 2N). the resulting matrix is X. We compute an N x N covariance matrix Ψ of individuals, where Ψjj´is defined to be the covariance of column j and column j´ of X. The kth axis of variation is the kth eigenvector of Ψ. The axes of variation reduce the data to a small number of dimensions, describing as much variability as possible; they are defined as the top eigenvectors of a covariance matrix between samples. This method allows the detection and correction of population stratification on a genome-wide scale, maximizing detection of true associations and minimizing erroneous associations [20].”
L173-176: This information doesn't seem relevant to me in that context. I'm not that much interested in the possibilities SVS offers but more in what was used in this study.
The box plot was requested by the other reviewer
L177: It is still not clear to me how the PCA was corrected for, it must be somewhere between equation 1 and here. Was there another step between equation 1 and 2 or were the PCAs included in eq 2?
“Where êi is the vector of residuals from (1), μ is the mean, g is the vector of markers corrected by PCA, k is the vector of markers effects, and e is the vector of random residuals.”
L206: In the description of Figure 1 please add the a and b names to make clear which plot is what
DONE
Figure 3: It should be mentioned somewhere that what is named 0 are the MT SNPs
DONE

Reviewer 2 Report
Several of my comments from the initial review have been addressed. However, I still find that the issues regarding power and the low statistical significance threshold have not been adequately explained.
1. In response to my comment about the low significance threshold, I do not think it is sufficient to say that the threshold was lowered to 0.001 simply due to the small sample size. There needs to be a better justification for this lowered level, either through the citation of a previous cattle GWAS that used this threshold or a valid genetic, biological, or mathematical reason.
2. Similarly, it is not enough when discussing the issues of power within this study to simply comment on the lowered significance threshold. There really needs to be a full paragraph in the Discussion that touches on the issues that an underpowered study can have, for instance potential spurious associations due to a lack of sample size, potential differences in allele frequency in the data set to the general population, etc. In general, the low power of this study is its major weakness and must be sufficiently addressed.
Author Response
Answer to the Academic Editor Notes.
The authors appreciate the comments. In general all the comments has been addressed.
REVIEWER 2.
Several of my comments from the initial review have been addressed. However, I still find that the issues regarding power and the low statistical significance threshold have not been adequately explained.
In response to my comment about the low significance threshold, I do not think it is sufficient to say that the threshold was lowered to 0.001 simply due to the small sample size. There needs to be a better justification for this lowered level, either through the citation of a previous cattle GWAS that used this threshold or a valid genetic, biological, or mathematical reason. Similarly, it is not enough when discussing the issues of power within this study to simply comment on the lowered significance threshold. There really needs to be a full paragraph in the Discussion that touches on the issues that an underpowered study can have, for instance potential spurious associations due to a lack of sample size, potential differences in allele frequency in the data set to the general population, etc. In general, the low power of this study is its major weakness and must be sufficiently addressed.
“Correcting the genotype through the PCA we avoid the genomic inflation factor. The genomic inflation factor expresses the deviation of the distribution of the observed test statistic compared to the distribution of the expected test statistic. High genomic inflation factors are caused by population stratification, strong linkage disequilibrium (LD) between SNPs, strong association between SNPs and phenotypes, and systematic bias [32].
As trait complexity increases, the number of loci affecting the trait increases along with environmental interactions with an expected decrease in heritability. Conversely, for complex traits, higher loci affect the trait, there is more interaction with the environment, and there is an expected decrease in heritability. For a trait with a low heritability, the threshold value for significance of associating loci with a trait would have low – log10 (P)-values [33]. In published studies in pig populations, threshold values for −log10 (p‐value) ranged from 3.3 to 6, using either no multiple testing correction, a Bonferroni correction, the false discovery rate, or genomic control [32]”

Round 3
Reviewer 1 Report
The authors have addressed now all major issues. Now a couple of minor issues that I still think should be adressed are given below:
The discussion on the power issue is still quite superficial, please go into more detail here, also the qq-plot seems to show no stratification but the deviation seems also quite low, such that the power issue becomes even more important.
L39: I wouldn't think that "increase in demand ... has stimulated ... reduce environmental costs" -> this might be more a social/ecologic question (which is also more in line with the arguments following afterwards)
L67 Without reading through ref. 4 from the this focus on breeding strategies, so there should also be references for the other strategies.
L68: Research has described -> sounds a little bit strange to me
L86: vast majority of cattle -> Please add a reference
L92: genomic regions associated with CH4 emissions
L108-109: What are DM, CP, NDF, ADF
L127: How was the average weight determined?
L131: There is no information on how often each cow was measured. Was this only once for each of the 280 cows?
L152: The MAF was determined across all samples or within group? It would be interesting to see the MAF per population for the significantly associated SNPs in Table S1
L157: recovery rate = call rate?
L158: I'd suggest to exclude the mitochonrial SNPs, in my opnion they only add confusion
L168/169: Hi and Mi are both fixed effects?
L169: ==
L170: You added subscript k to the residual, this should also be added in the description
L171: ... were also evaluated in the model design ?
L180: bracket closing missing
L177-L187: The explanation of how the PCA correction was done is now quite long and confusing: where does the kth axis in L183 come from? How many eigenvectors were used for the correction and I still don't see how they are then integrated in Ψ. My understanding: You obtained the eigenvectors of the Gmatrix and then you corrected the vector of genotypes by these eigenvectors?
L198: subscript i is the ith SNP?
L194: How are the residuals corrected for polygenic covariation? Aren't they just corrected for the effect of H and M? And the g would be corrected for the polygenic covariance?
L229: Delete "Supported by the PCA, the data were corrected of population stratification on a genome-wide scale."
L244: The statement that all associated markers are common in the population I'm not sure about: If you'd increase your MAF threshold to 10% you'd loose about half of your associated SNPs. For this reason I'dlike to see the MAF per population, in order to verify, that all of these markers segregate in all populations (maybe per system or preferably per PCA cluster).
L276: You mean e.g. "we avoid genomic inflation" or "we avoid a high genomic inflation factor" ?
L283: higher number of loci
L286: As the argumentation is the heritability of the trait the heritability of the trait in the cited study and the estimated heritability for methane emission should be given.
L289: It would be something like: SNPs identified to be associated to methane emission in this study on ... were in other studies associated with ...
L296-305: Please say for every QTL you describe, for which trait the QTL is, and where you mean the SNPs and QTL you identified in this study.
L308: This study found gene regions associated to intramuscular fat content -> this study only found gene regions associated with methane emissions
L314 residual feed intake
L353-354: The description of the Supplementary data is missing here. Also it would be nice if supplementary figures S1-S8 could be combined in a single figure or at least a single file. Furthermore, by just looking at these figures the description here is to sparse. It is unclear what "QTL" means in that context. It should be described for each figure (in its legend), for what trait the QTL (if I got that right these are the QTL from QTLdb, or are they from somewhere else?) is, it should also be explained what all the colored flags mean.
L361: The funding and acknowledgement sections are also still empty
Generally gene names should be written in italics.
I'd recommend to go through the whole discussion again and check if all QTLs are described properly and it is clear what information came from where.
Author Response
The authors appreciate the comments made and have allowed us to have a better version of our manuscript.
L39: I wouldn't think that "increase in demand ... has stimulated ... reduce environmental costs" -> this might be more a social/ecologic question (which is also more in line with the arguments following afterwards)
Authors: The sentence has been revised and the suggestion has been incorporated.
L67 Without reading through ref. 4 from the this focus on breeding strategies, so there should also be references for the other strategies.
Authors: References # 4 is Intitule Developing breeding schemes to assist mitigation of greenhouse gas emissions. This reference cover all the strategies mentioned in the sentence.
L68: Research has described -> sounds a little bit strange to me
Authors. Line 68 considers it is right.
L86: vast majority of cattle -> Please add a reference
Authors. “Vast” as deleted. The sentence remains as follow: “even though these countries account for the majority of cattle in the world”
L92: genomic regions associated with CH4 emissions
Authors: The suggestion was incorporated.
L108-109: What are DM, CP, NDF, ADF
Authors: Definitions have been incorporated. Dry Matter (DM), Crude Protein (CP), Neutral Detergent Fiber (NDF), Acid Detergent Fiber (ADF).
L127: How was the average weight determined?
Authors. The weight was estimated by using tape measure. The sentence e is as follow: Using tape measure, the average weight of the cows was 546 kg
L131: There is no information on how often each cow was measured. Was this only once for each of the 280 cows?
Authors. This information has been incorporated. As follow “During milking and for six weeks2
L152: The MAF was determined across all samples or within group? It would be interesting to see the MAF per population for the significantly associated SNPs in Table S1
Authors. MAF was estimated across all the samples. MAF is a measure of the frequency of the SNP. At this stage, the significance of the association is more informative for the purpose of the study. Differences in the frequencies of the SNP could be interesting to other aims of this project.
L157: recovery rate = call rate?
Authors. Done.
L158: I'd suggest to exclude the mitochondrial SNPs, in my opinion they only add confusion
Authors. We considers useful to include the mitochondrial SNPs, even just one m-snp was associated.
L168/169: Hi and Mi are both fixed effects?
Authors: Both are fixed effect. Incorporated into the model description.
L169: ==
Authors. Deleted.
L170: You added subscript k to the residual, this should also be added in the description
Authors: Done.
L171: ... were also evaluated in the model design?
Authors. The authors evaluated the model thought significance, r-squared and CME.
L180: bracket closing missing
Authors: Done.
L177-L187: The explanation of how the PCA correction was done is now quite long and confusing: where does the kth axis in L183 come from? How many eigenvectors were used for the correction and I still don't see how they are then integrated in Ψ. My understanding: You obtained the eigenvectors of the Gmatrix and then you corrected the vector of genotypes by these eigenvectors?
Authors: We consider that the PCA corrections is not the principal aim of this study. We just apply the methodology proposed in the literature. Please for further details review Price, A.L.; Patterson, N.J.; Plenge, R.M.; Weinblatt, M.E.; Shadick, N.A.; Reich, D. Principal components analysis corrects for stratification in genome-wide association studies. Nat. Genet. 2006, 38, 904–909 and Golden Helix SNP Genome-Wide Association Tutorial; 2017;
L198: subscript i is the ith SNP?
Authors: Yes.
L194: How are the residuals corrected for polygenic covariation? Aren't they just corrected for the effect of H and M? And the g would be corrected for the polygenic covariance?
Authors. Authors consider here is a misunderstanding. The residuals were not corrected by polygenic variation. H and M remove the environmental variation. Authors consider that the follow sentence explain properly the idea, “The effects included in the model explain a part of the environmental variation. The residuals represent the proportion of the variance not explained by the model effects, including genetic variance [19], for that reason residuals values were used as phenotypes in the GWAS”.
L229: Delete "Supported by the PCA, the data were corrected of population stratification on a genome-wide scale."
Authors. Correction done.
L244: The statement that all associated markers are common in the population I'm not sure about: If you'd increase your MAF threshold to 10% you'd loose about half of your associated SNPs. For this reason I'dlike to see the MAF per population, in order to verify, that all of these markers segregate in all populations (maybe per system or preferably per PCA cluster).
Authors. MAF was estimated across all the samples. MAF is a measure of the frequency of the SNP. At this stage, the significance of the association is more informative for the purpose of the study. Differences in the frequencies of the SNP could be interesting to other aims of this project.
L276: You mean e.g. "we avoid genomic inflation" or "we avoid a high genomic inflation factor" ?
Authors. Correction done.
L283: higher number of loci
Authors.
L286: As the argumentation is the heritability of the trait the heritability of the trait in the cited study and the estimated heritability for methane emission should be given.
Authors. The heritability of the methane emissions Reference number 13. In this study this was not an aim for this study.
L289: It would be something like: SNPs identified to be associated to methane emission in this study on ... were in other studies associated with ...
Authors. Suggestion incorporated.
L296-305: Please say for every QTL you describe, for which trait the QTL is, and where you mean the SNPs and QTL you identified in this study.
Authors. Correction done.
L308: This study found gene regions associated to intramuscular fat content -> this study only found gene regions associated with methane emissions
Authors. Suggestion incorporated.
L314 residual feed intake
Authors. Suggestion incorporated.
L353-354: The description of the Supplementary data is missing here. Also it would be nice if supplementary figures S1-S8 could be combined in a single figure or at least a single file. Furthermore, by just looking at these figures the description here is to sparse. It is unclear what "QTL" means in that context. It should be described for each figure (in its legend), for what trait the QTL (if I got that right these are the QTL from QTLdb, or are they from somewhere else?) is, it should also be explained what all the colored flags mean.
Authors. Suggestion incorporated.
L361: The funding and acknowledgement sections are also still empty
Authors. Information incorporated.
Generally, gene names should be written in italics.
Authors: Done.
I'd recommend to go through the whole discussion again and check if all QTLs are described properly and it is clear what information came from where.

Reviewer 2 Report
I feel that all my comments have been sufficiently addressed
Author Response
The authors appreciate the comments made and have allowed us to have a better version of our manuscript.